# Structural and mechanistic insights into a lysosomal membrane enzyme HGSNAT involved in Sanfilippo syndrome

Boyang Zhao[1,8], Zhongzheng Cao[2,8], Yi Zheng [3], Phuong Nguyen[4,6], Alisa Bowen [4,7], Robert H. Edwards [5], Robert M. Stroud [4], Yi Zhou [2], Menno Van Lookeren Campagne [2] & Fei Li [1]✉

Heparan sulfate (HS) is degraded in lysosome by a series of glycosidases. Before the glycosidases can act, the terminal glucosamine of HS must be acetylated by the integral lysosomal membrane enzyme heparan-α-glucosaminide *N*-acetyltransferase (HGSNAT). Mutations of HGSNAT cause HS accumulation and consequently mucopolysaccharidosis IIIC, a devastating lysosomal storage disease characterized by progressive neurological deterioration and early death where no treatment is available. HGSNAT catalyzes a unique transmembrane acetylation reaction where the acetyl group of cytosolic acetyl-CoA is transported across the lysosomal membrane and attached to HS in one reaction. However, the reaction mechanism remains elusive. Here we report six cryo-EM structures of HGSNAT along the reaction pathway. These structures reveal a dimer arrangement and a unique structural fold, which enables the elucidation of the reaction mechanism. We find that a central pore within each monomer traverses the membrane and controls access of cytosolic acetyl-CoA to the active site at its luminal mouth where glucosamine binds. A histidine-aspartic acid catalytic dyad catalyzes the transfer reaction via a ternary complex mechanism. Furthermore, the structures allow the mapping of disease-causing variants and reveal their potential impact on the function, thus creating a framework to guide structure-based drug discovery efforts.

Heparan sulfate (HS) is a glycosaminoglycan polysaccharide found in proteoglycans associated with the cell membrane and extracellular matrix in nearly all cells. It plays critical roles in mediating intercellular and intracellular interactions that underlie a plethora of functions such as cell recognition, adhesion, and mobility. Thus, it exerts a profound influence on normal physiology as well as on cancer, and metabolic, inflammatory, and neurodegenerative diseases[1]. HS is a linear polysaccharide composed of repeating disaccharide units of glucosamine (GlcN) and uronic acid with varying degrees of modifications[2]. Degradation of HS starts on the cell surface and ends in the lysosome, catalyzed by a series of enzymes[3]. The lysosomal membrane protein, heparan sulfate acetyl-CoA:α-glucosaminide *N*-acetyltransferase (HGSNAT), also known as TMEM76, plays the critical role of acetylating the terminal GlcN[4,5]. As the only non-catabolic enzyme in the lysosome,

[1]Amgen Research, Department of Structural biology, South San Francisco, CA, USA. [2]Amgen Research, Department of Inflammation, South San Francisco, CA, USA. [3]Amgen Research, Department of Discovery Protein Science, South San Francisco, CA, USA. [4]Department of Biochemistry and Biophysics, University of California San Francisco (UCSF) School of Medicine, San Francisco, CA, USA. [5]Departments of Neurology and Physiology, UCSF School of Medicine, San Francisco, CA, USA. [6]Present address: Laboratory for Genomics Research, UCSF School of Medicine, San Francisco, CA, USA. [7]Present address: Adanate, Alameda, CA, USA. [8]These authors contributed equally: Boyang Zhao, Zhongzheng Cao. ✉e-mail: fli05@amgen.com

the proper function of HGSNAT is essential for HS degradation, as none of the lysosomal glycosidases can act on the unacetylated GlcN[3]. Indeed, mutation of HGSNAT leads to lysosomal accumulation of HS and a severe form of lysosomal storage disease (LSD) termed mucopolysaccharidosis IIIC (MPS IIIC), or Sanfilippo syndrome C, for which no treatment is available[6].

HGSNAT is ubiquitously expressed (https://www.proteinatlas.org/ENSG00000165102-HGSNAT) and highly conserved among all species (Supplementary Fig. 1) but has little similarity to other proteins of known functions. It catalyzes a unique transmembrane acetylation reaction where the acetyl group of cytoplasmic acetyl-CoA is transferred to HS in the lysosomal lumen in one reaction (EC:2.3.1.78) (Fig. 1a). Despite being identified as the critical enzyme for lysosomal degradation of HS and the causative gene for MPS IIIC since 1978[4,5,7], the reaction mechanism of HGSNAT remains contentious[8–11], due largely to the protein's unique structural fold and lack of any structural information until very recently[12]. Critical questions such as (1) how HGSNAT catalyzes the synthetic reaction involving acetyl-CoA in the lysosomal environment, (2) how HGSNAT combines the functions of both transporter and enzyme within one protein, and (3) if the reaction goes through a single step or multiple steps, remain unresolved.

In this work, we determine 6 high-resolution structures of human HGSNAT in different states along the reaction pathway by cryo-electron microscopy (cryo-EM). The results reveal a dimeric arrangement of HGSNAT with a central substrate binding pore traversing the entire membrane, which allows the transmembrane acetylation of lysosomal HS by cytosolic acetyl-CoA in one single reaction. The structures not only allow us to determine the reaction mechanism at the atomic level but also provide critical insight for the disease-causing variants and serve as the basis for the rational design of therapeutics.

## Results

### Structure determination of human HGSNAT

Purified HGSNAT shows robust activity and homogenous behavior suitable for cryo-EM studies (Supplementary Fig. 2). We observed 2 main species of HGSNAT in all samples (Supplementary Fig. 2d, f), corresponding to the full-length protein and the auto-cleaved form (HGSNAT-β) as previously reported[9,11]. The 2 species elute in a single

homogenous peak on size exclusion chromatography, and we did not observe any structure representing the cleaved HGSNAT-β where only the transmembrane domains are present. Our results thus confirm that the cleaved N-terminal portion (HGSNAT-α) remains associated with the C-terminal HGSNAT-β.

To gain insights into the molecular mechanism of HGSNAT function, we determined 6 high-resolution structures of HGSNAT along the reaction pathway by cryo-EM (Supplementary Fig. 3, table 1): two apo structures in different conformations where (1) apo protein is in the ground state at 3.49 Å [Apo$_{ground}$] and (2) apo protein in transition state at 3.61 Å [Apo$_{trans}$]; (3) HGSNAT bound with acetyl-CoA at 2.92 Å [Acetyl-CoA]; (4) HGSNAT bound with acetyl-CoA and a substrate analog 4-methylumbelliferyl-$\beta$-d-glucosaminide (MU-$\beta$GlcN)[11] at 3.15 Å [substrates complex]; (5) HGSNAT bound with CoA and a product analog 4-methylumbelliferyl-$N$-acetyl-$\beta$-d-glucosaminide (MUF-NAG)[11] [products complex] at 3.20 Å; and (6) HGSNAT bound with CoA at 3.12 Å [CoA]. Aside from being an integral part of the reaction cycle, the lack of extra density representing the acetyl group in the [CoA] structure compared to the [Acetyl-CoA] structure helped us to precisely model the position of the Acetyl group. Comparison of all 4 ligand bound structures helped us to better understand the acetyl transfer reaction.

### Overall architecture of HGSNAT

HGSNAT forms a dimer perpendicular to the membrane with dimensions of 90 Å x 90 Å x 60 Å. All interactions between the monomers are located within the transmembrane domain (TMD), whereas the two extracellular domains (ECD) are located at opposite sides of the dimer (Fig. 1b, d, e). HGSNAT forms a negatively charged cavity on the lumen side, while the cytosolic surface is positively charged (Fig. 1c). The negative charge on the luminal cavity might help reduce non-specific ionic interaction with the highly negatively charged HS so that it can be precisely positioned for catalysis. Each monomer of HGSNAT is composed of an N-terminal ECD followed by a long tilted TM1 and a compact C-terminal TMD formed by TM2-TM11 (Fig. 1d, e). The first 75 residues and the majority of the intracellular loop 1 (ICL1) connecting TM1 and TMD are unresolved, suggesting that these regions are unstructured. The ECD forms a $\beta$-sandwich structure, typical for

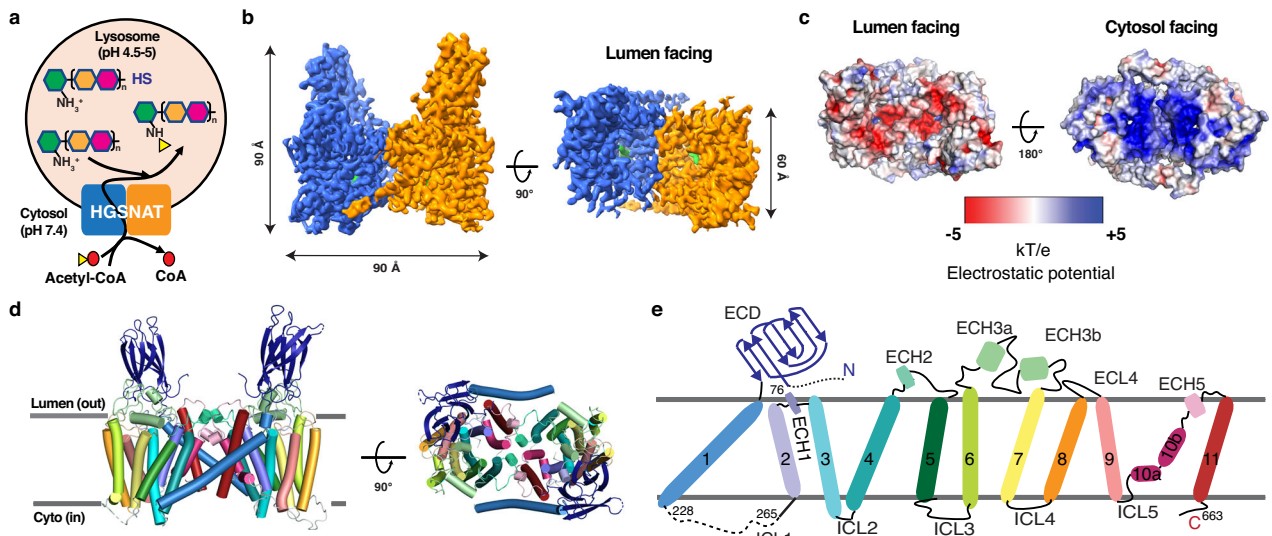

**Fig. 1 | Function and overall structure of human HGSNAT. a** HGSNAT catalyzes the transmembrane acetylation of HS in the lysosome. The disaccharide repeating unit is represented by orange and magenta hexagons, while the terminal GlcN is shown as a green hexagon. Acetyl-CoA is represented by a yellow triangle (acetyl group) connected with a red circle (CoA). **b** Cryo-EM structure of human HGSNAT bound with acetyl-CoA. The two monomers within the dimer are colored in blue and orange respectively. Acetyl-CoA is colored in green. **c** Electrostatic potential of HGSNAT dimer. **d** Structure of HSGNAT dimer. **e** Topology map of the HGSNAT monomer colored in discrete colors corresponding to (**d**).

carbohydrate-binding modules (CBMs)[13]. Structural comparison using Foldseek[14] identifies that the ECD resembles CBM20 family proteins, which are type B CBMs that bind to internal glycan chains[15], suggesting that the function of the ECDs might be to bind and position the extended HS chain for optimal catalysis. Within the TMD, helices towards the center of the helical bundle exhibit more tilting. Particularly, TM10 at the center is broken into 2 segments with half of the helix (10a) parallel to the membrane while the other half (10b) only spanning ~15 Å in the middle of the membrane. The extracellular loop (ECL) regions contain 5 short helical segments (ECH1-5) that interact with each other extensively and form the luminal cavity on top of the TMD. The long ECL3 as well as ECL1 and ECL4 also interact with the ECD, while ECL2 and ECL5 interact with each other reciprocally and form the dimer interface (Fig. 1d, e).

## Substrates binding sites and the reaction mechanism

The substrates binding sites of HGSNAT had not been well defined, while the mechanism that allows it to catalyze the synthetic reaction in the lysosome has been debated for decades[8–11]. To address these questions, we determined structures of both the substrates and products complexes (Fig. 2a–c and Supplementary Fig. 4a). Acetyl-CoA binds in a linear fashion that traverses the entire TMD, while MU-$\beta$GlcN binds at the luminal mouth of the acetyl-CoA binding site with the amine group facing the acetyl group on acetyl-CoA (Fig. 2a, b). The CoA moiety interacts with HGSNAT through an extensive network of polar and hydrophobic interactions: a large number of polar interactions at the cytosolic mouth stabilize the 3'-phosphoadenosine, while R345 and R275 form hydrogen/ionic bonds with the diphosphate; the pantothenate group is accommodated through hydrophobic interactions, while the cysteamine forms a hydrogen bond with M282. The carbonyl of the acetyl group is stabilized by N286 (Fig. 2d), whose mutation is identified in MPS patients[16]. The GlcN moiety's interaction with the luminal cavity formed by the ECLs is mediated mostly by polar interactions (Fig. 2d), including with the proposed catalytic residue H297[11,17]. R372 and E499, whose mutations lead to low activity and mislocalization of HGSNAT in MPS patients[18–20], bind to the 6'-OH and 3'-OH of GlcN respectively.

A histidine residue has been proposed to be the catalytic residue since the 1980s[17] and was confirmed through mutagenesis after the gene was cloned[11]. Two mechanisms involving a catalytic histidine have been proposed for the acetylation reaction by HGSNAT, namely the ternary complex mechanism and the ping-pong mechanism (Fig. 2e). In the ternary complex mechanism, both substrates are expected to bind to HGSNAT at the same time to form a ternary complex with the enzyme, which allows the direct transfer of the acetyl group from acetyl-CoA to GlcN. On the other hand, the two substrates would bind to HGSNAT sequentially in the ping-pong mechanism, which requires the formation of the acetylated protein as a reaction intermediate. Despite detailed biochemical characterizations, the reaction mechanism remained elusive as the acidic environment within the lysosome would prohibit the histidine from acting as the nucleophile in both proposed mechanisms. We identified a catalytic dyad of H297-D307 in the structures (Fig. 2f, g), which lowers the p$K_a$ of H297 to ~4, as calculated by PROPKA[21]. Meanwhile, H297 is positioned within a distance that would allow it to deprotonate the amine of GlcN (Fig. 2f). The deprotonated GlcN would then carry out a nucleophilic attack at the acetyl carbonyl group, while the leaving group CoA removes the proton from H297, regenerating the active site via a ternary complex mechanism (Fig. 2h). In addition, N286 is positioned 2.7 Å away from the carbonyl oxygen of the acetyl group, which stabilizes the negative charge in the intermediate state (Fig. 2f, h). Consistent with the ternary complex mechanism, kinetic studies by *Meikle* et. al. observed converging sets of lines in the Lineweaver-Burk plots[8]. On the contrary, $N_\varepsilon$ of H297 is further away from the acetyl group compared to the amine of GlcN (Fig. 2f). While H297 could attack acetyl-CoA, the positively charged GlcN would not be able to attack the electrophilic carbonyl on the resulting intermediate or the acetylated H297 (Fig. 2i), leaving the acetylated protein as a dead-end product rather than an intermediate proposed for the ping-pong mechanism. Consistent with this, the dead-end product is observed only in the absence of GlcN and can be reversed by the addition of CoA[10]. Additionally, the reported complete inhibition of acetylation of HGSNAT in the presence of GlcN[10] is more consistent with the ternary complex mechanism while only partial inhibition would be expected if GlcN is reacting with the intermediate in the second step of the reaction. Consistent with their critical roles, H297, D307, as well as N286, are highly conserved (Fig. 2j). Mutations of H297, D307, and N286 indeed drastically reduced the activity of HGSNAT (Fig. 2k and Supplementary Fig. 5). While all mutations reduced the activity, mutations of H297 caused the most severe effects, consistent with it being the catalytic residue. The very low activity of H297D suggests that while the charge state of H297 is important for the nucleophilic reaction, the geometry of the active site is also critical. Similarly, the N286D mutation has a much milder influence on the activity compared to the N286Q and N286A mutations. The mutation of D307 also reduced the activity significantly but less severely compared to the H297 mutations. In addition, H297D/D307N was not able to rescue the defect of H297D. Taken together, the mutagenesis data confirms the importance of H297, D307, and N286 while also underlining the importance of the active site geometry.

## Substrate is not transported across the membrane

Unlike other glycan modification enzymes that require a separate transporter to transport substrates cross the membrane[22–24], HGSNAT fulfills both functions in one protein. It is thus proposed that HGSNAT itself cycles between cytosolic-open and luminal-open conformations to transport acetyl-CoA as the first step of the transmembrane acetylation reaction[17,25]. However, we observed no major conformational change of the TMD between different states (Supplementary Fig. 6a). Rather, the acetyl-CoA binding site forms a continuous positively charged pore as the binding pocket that allows it to transverse the TMD and position the acetyl group at the active site at the luminal mouth (Fig. 3a). Accessibility to the active site is gated by subtle sidechain conformational changes of a few pore-lining residues (Fig. 3a–c and Supplementary Fig. 6b, c). Sidechain rotations of M282, F310, F313 on TM2 and TM3 as well as I608 and Y611 on TM10b completely block the pore in the Apo$_{ground}$ structure and render it significantly restricted in the Apo$_{trans}$ structure. The cytosolic mouth of the pore also exhibits notable structural rearrangements (Fig. 3d, e). Particularly, ICL1, where the adenine binding residue R267 is located, moves away from the pore in the Apo structures. ICL1 connects the TMD with the highly flexible TM1 (Fig. 3f and Supplementary Figs. 7, 8). The first half of TM1 is completely invisible in the Apo$_{trans}$ structure while a kink around I196 leads to an approximate 29° rotation between the Apo$_{ground}$ and all ligand(s) bound structures (Fig. 3f and Supplementary Fig. 7b, c). We also compared our structures with the Alpha-Fold prediction and the acetyl-CoA bound structure of HGSNAT recently determined by *Navratna* et al. while our paper is in revision (8TU9)[12] (Supplementary Fig. 8). TM1 is predicted to be a broken helix with a kink around L217 by AlphaFold. A shorter portion of TM1 (till residue F210) is resolved in 8TU9 in a conformation similar to that predicted by the AlphaFold but drastically different from what we observed for the acetyl-CoA bound structure. While it is unclear if the N-terminal GFP tag present in the 8TU9 structure and the lack of cleavage in ICL1 had any influence on the conformation of TM1 or if this conformation is physiological, the collection of the structures clearly demonstrates that TM1 is highly dynamic. Conformational changes and dynamics of the TM1 and ICL1 could therefore influence the accessibility of acetyl-CoA from the cytosol and play a role in gating.

The dimeric arrangement of HGSNAT does exhibit significant dynamics between the 6 different states due to structural changes

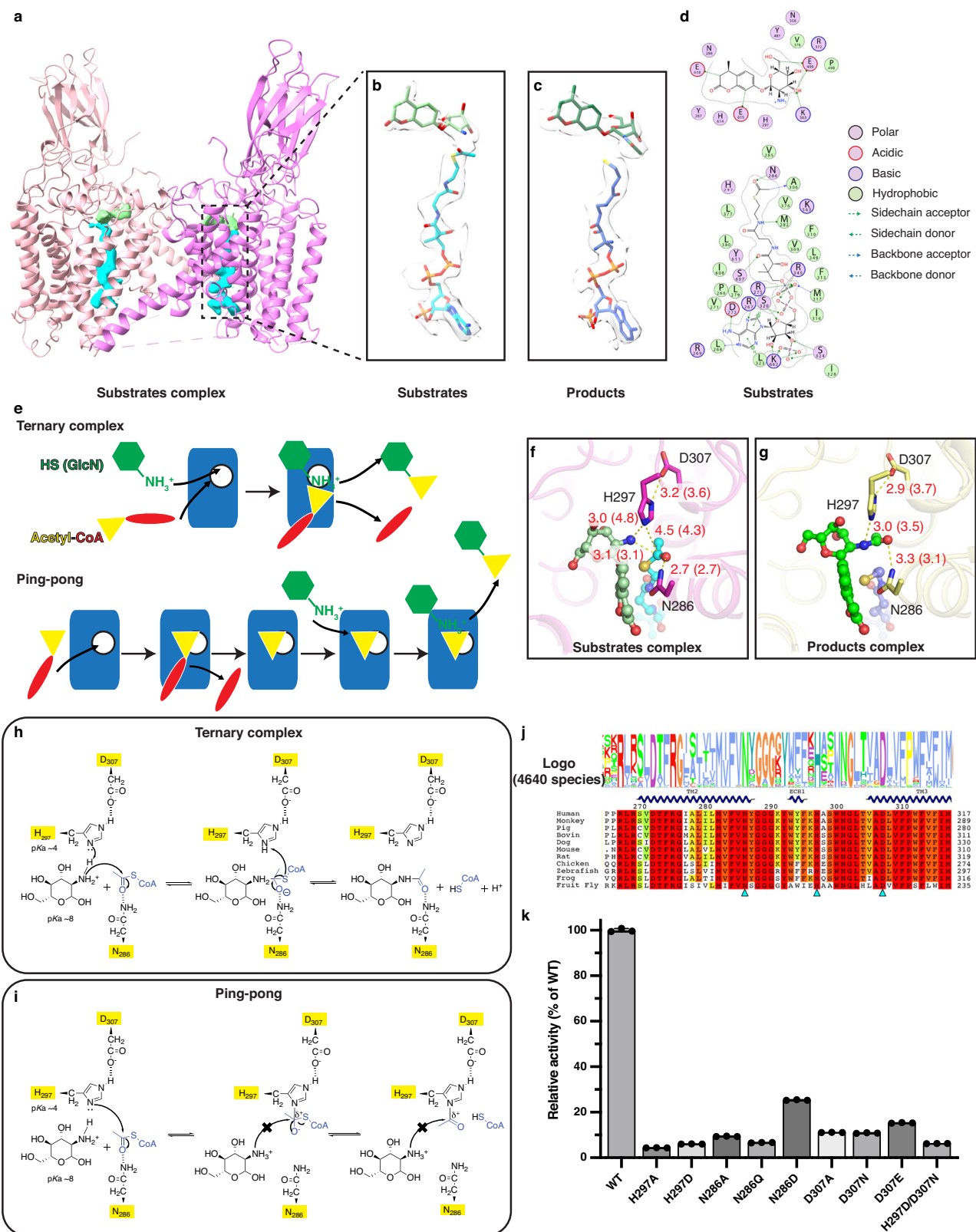

at the dimer interface (Fig. 3g–j and Supplementary Fig. 8f). The ligand(s) bound structures as well as the Apo$_{trans}$ structure adopt similar structures with a dilated configuration where Y361 folds back into the interface and interactions are mostly hydrophobic (Fig. 3h, i). On the other hand, the Apo$_{ground}$ structure adopts a constricted configuration where the monomers interact at both the luminal and the cytosolic surfaces (Fig. 3j). Conformational change

of ECL2 between I356 and R372 results in a different dimer interface with fewer hydrophobic interactions (Fig. 3j and Supplementary Fig. 6a). C362 is located at the dimer interface. However, contrary to a previous hypothesis[11], disulfide does not play a critical role in the dimer formation. C362 of both monomers face each other in a conformation compatible with disulfide bond formation in the Apo$_{trans}$ and all ligand(s) bound structures. However, we can only

**Fig. 2 | Substrates binding and reaction mechanism. a** Structure of the substrates complex with the densities of acetyl-CoA in cyan and MU-*β*GlcN in light green. **b** Densities and models of acetyl-CoA (cyan) and MU-*β*GlcN (light green) in monomer B of the substrates complex. **c** Densities and models of CoA (blue) and MUF-NAG (green) in monomer B of the products complex. **d** 2D ligand interaction plot for the substrates binding site. **e** Schematic of the ternary complex and ping-pong mechanisms. A monomer of HGSNAT is represented by the blue rectangle with the active site shown as white circle. **f** Structure of the substrates binding site. **g** Structure of the product binding site. Interactions between atoms are shown by dotted lines and distances (in Angstroms) between atoms involved are reported in red. The structure of only one monomer is shown but distances between the equivalent residues for the other monomer are labeled in parenthesis. **h** Ternary complex mechanism. **i** Ping-pong mechanism. **j** Logo plot (in Clustal 2 color scheme) of 4640 species and sequence alignment of 11 representative species in the region where N286, H297, and D307 are located. **k** Enzymatic activities of HGSNAT mutants. All the enzymatic activities of HGSNAT mutants were normalized to the expression level and shown as % of HGSNAT WT. Results are shown as mean ± SD (*n* = 3 replicates). Source data are provided as a Source Data file.

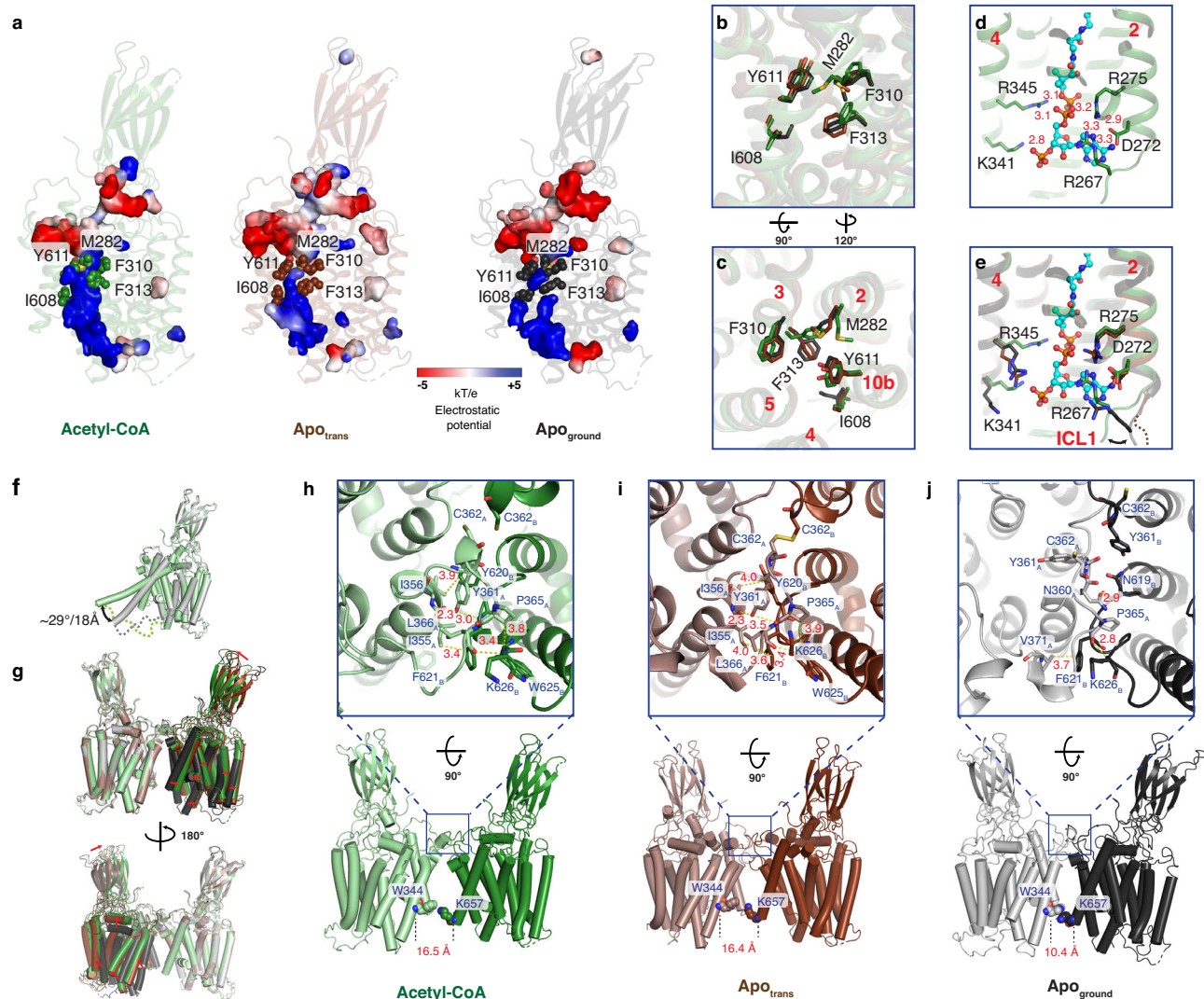

**Fig. 3 | Protein dynamics influence acetyl-CoA accessibility. a** Acetyl-CoA binding pore within TMD shown in surface representation with the electrostatic potential. Only one monomer is shown for simplicity. **b, c** Sidechain conformation changes gate the acetyl-CoA binding pore. Residues from the Acetyl-CoA structure are colored in green, residues from the Apo_trans structure are colored in brown, and residues from the Apo_ground structures are colored in black. **d** Cytosolic portion of the acetyl-CoA binding pore in the Acetyl-CoA structure. **e** Overlay of the cytosolic portion of the acetyl-CoA binding pore in the Acetyl-CoA structure (green), Apo_trans structure (brown) and the Apo_ground structure (black). **f** Overlay of monomers of the Acetyl-CoA structure (light green) and the Apo_ground structure (gray) shows a significant conformation change of TM1. **g** Overlay of the dimers of the Acetyl-CoA structure (green/light green), Apo_trans structure (brown/light brown), and the Apo_ground structure (black/gray) shows a large rotation movement. TM1 is hide for clarity. **h–j** Dimer interface of the Acetyl-CoA structure (**h**), the Apo_trans structure (**i**), and the Apo_ground structure (**j**). Note, all ligand(s) bound structures exhibit similar conformation as the Acetyl-CoA structure (Supplementary Figs. 6, 7b). For simplicity, Acetyl-CoA is not shown in the Acetyl-CoA structure except in panel d and e.

observe a disulfide bond convincingly in the Apo_trans structure but not in the presence of ligand(s) (Fig. 3h, i and Supplementary Fig. 4b–e). On the other hand, both C362 residues point into the lumen in a configuration incompatible with disulfide formation in the Apo_ground structure (Fig. 3j and Supplementary Fig. 4f, g). The

observations of different dimer arrangements between all 6 structures and the breaking/formation of disulfide bond at the dimer interface suggest that the intrinsic dynamics of the dimer contribute to the activation of HGSNAT, which precedes the opening of the acetyl-CoA binding pore.

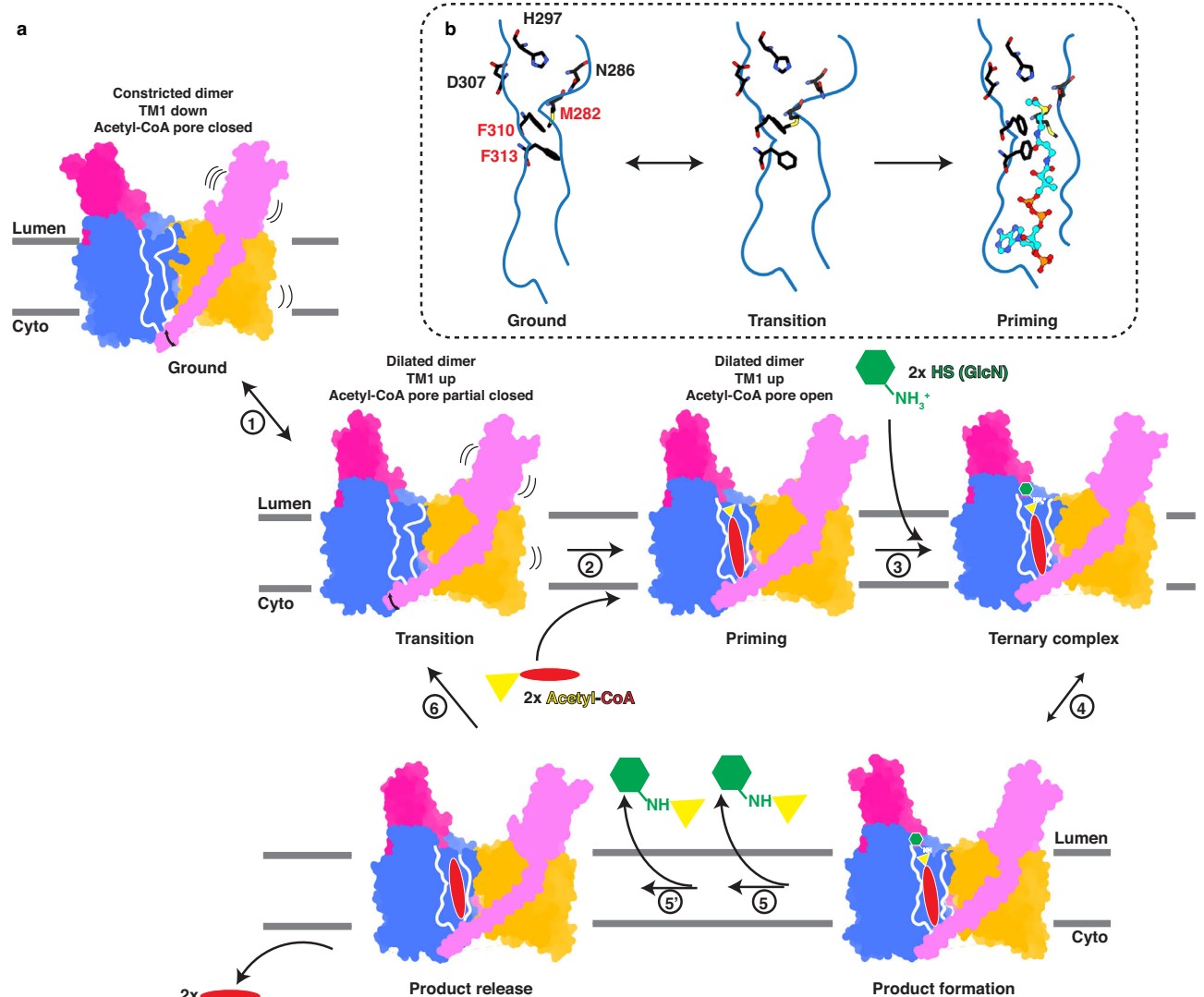

**Fig. 4 | Model for the transmembrane acetylation reaction catalyzed by HGSNAT. a** Structural model of the reaction cycle of HGSNAT (steps 1–6). TMDs are colored in blue and orange while the ECDs and TM1s are colored in magenta and pink. The acetyl-CoA binding pore is represented by its outline. HS (GlcN) is shown as a green hexagon while acetyl-CoA is represented by a yellow triangle (acetyl group) connected with a red oval (CoA). HGSNAT exhibits an equilibrium between the ground state and transition state without substrate bound (step 1) and transitions into the priming state when acetyl-CoA binds (step 2). HS binding (step 3)

completes the catalytic ternary complex and allows the reaction to proceed (step 4). Subsequent release of both substrates (steps 5 and 6) allows HGSNAT to return to the equilibrium between ground and transition states. **b** Zoom in of the acetyl-CoA binding pore and the active site with critical residues shown in black stick and acetyl-CoA shown in cyan ball-and-stick. M282, F310, and F313, where conformational changes that gate the acetyl-CoA binding pore are observed, are highlighted in red. For simplicity, the reaction in only one monomer is illustrated.

## Structural model for the transmembrane acetylation reaction

Based on the structures, we propose the following model for the unique transmembrane acetylation reaction catalyzed by HGSNAT (Fig. 4). Without either substrate bound, the acetyl-CoA binding pore is closed but the HGSNAT dimer exhibits an equilibrium between the ground and the transition states where the dimer interface and TM1 display significant dynamics. Sidechain rotation of the gating residues allows slight dilation of the acetyl-CoA binding pore in the transition state. However, the pore remains closed. Acetyl-CoA binding then fully opens the pore and positions the acetyl group at the active site located at the luminal mouth of it in the priming state. This configuration of the substrates binding and the active site is the structural basis for the transmembrane acetylation reaction. The binding of the second substrate, HS, from the lumen completes the catalytic ternary complex and allows the direct transfer of the acetyl group to HS. While the binding order could reverse, with HS binding first, opening of the acetyl-CoA binding pore is likely still the rate limiting step in substrates

binding, as the active site is readily accessible from the lumen. We observed slightly different configurations of the active site between the two monomers where one monomer appears to have bonding distances better suited for catalysis than the other (Supplementary Fig. 9), suggesting that the two monomers may be in active state asynchronously. Consistent with this observation, two different $K_m$ values for both acetyl-CoA and GlcN are reported[8]. After releasing of all products, HGSNAT returns to the apo equilibrium.

## Structural basis for MPS IIIC disease

Approximately 28 missense variants of HGSNAT have been identified in patients with MPS IIIC[16,18]. A certain degree of defects in protein expression and trafficking are suggested in many cases[19,20]. However, no information on their specific influence on the structure and function of HGSNAT is available. To gain better understanding of their disease-causing mechanism and provide the basis for structure-based drug design, we mapped the variants on the structure of HGSNAT and

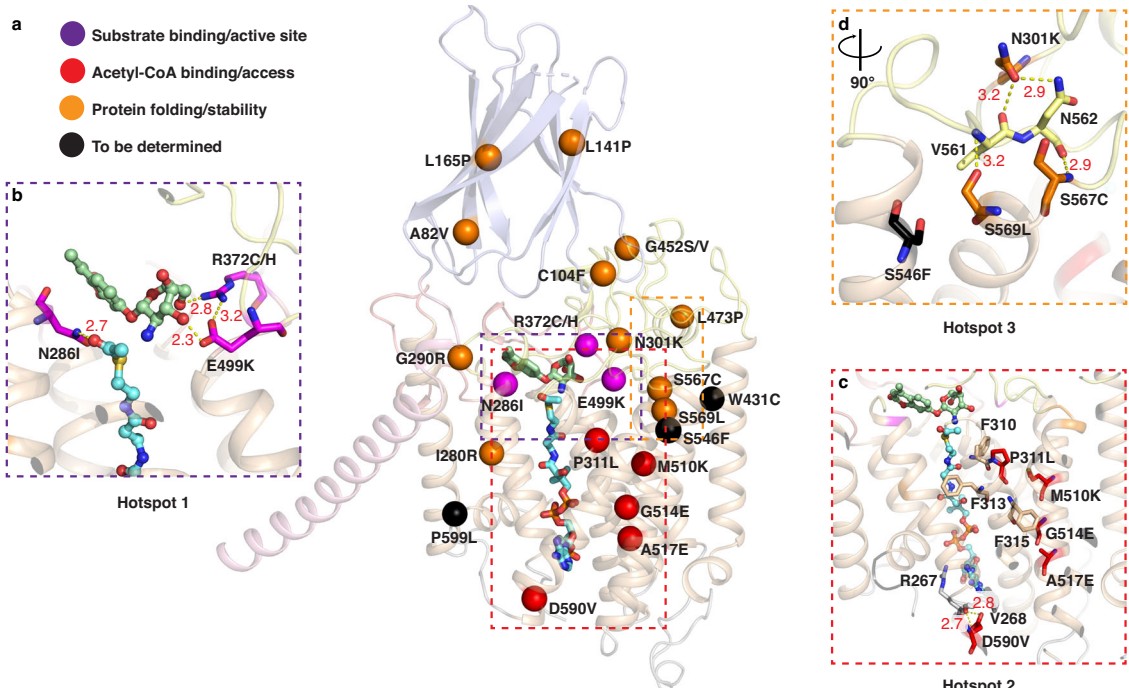

**Fig. 5 | MPS IIIC variants mapped on the structure of HGSNAT. a** MPS IIIC variants mapped on HGSNAT. ECD is colored in light blue and the TM1 is in light pink. TMD is colored in wheat while the ECL loops are colored in light yellow. **b** Zoom in of the mutation hotspot 1 where substrates binding is affected. **c** Zoom in on the mutation hotspot 2 which influences the acetyl-CoA accessibility. **d** Zoom in of the mutation hotspot 3 at the interface between TMD and ECL.

analyzed their implication on the structure and function (Fig. 5). R372C/H, E499K, and N286I are in a position to directly disrupt the binding of substrates GlcN and acetyl-CoA and resulted in reduced activity when tested in vitro[16,19] (Fig. 5b). P311L, M510K, G514E, A517E, and D590V could influence the accessibility of acetyl-CoA through disturbing critical residues that line the acetyl-CoA binding pore, such as F310, F313, F315, and R267 (Fig. 5c). Reversal of non-polar and polar residues with these mutations, especially within TMD, is likely to cause misfolding of the protein as well. A third hotspot for mutation is located at the interface between the TMD and the ECLs (Fig. 5d). Mutation of either N301K, S567C, or S569L will likely disrupt the interaction network with V561 and N562 and influence the folding/ stability of the protein. The ECLs appear to play an important role in protein folding/stability as two more mutations (G452S/V and C104F) are located at the interface between ECLs and ECD. Three more mutations (A82V, L185P, and L141P) are located on the ECD, where they could influence the folding and stability of the ECD and likely the binding of long-chain HS. Other mutations such as I280R and G290R, introduce large, charged residues within the TMD likely causing misfolding of the protein.

## Discussion

The high-resolution structures in multiple conformations determined in this study provide the opportunity to gain a comprehensive understanding of the function of HGSNAT. HGSNAT utilizes a canonical histidine-based mechanism to catalyze an acetyl transfer reaction. However, to adapt to the harsh environment in the lysosome[26], HGSNAT evolved a unique structural fold unobserved in any other proteins. The local structure around the catalytic H297 lowers its p$K_a$ extensively so that it could function as a nucleophile at extremely low pH in the lysosome. A continuous binding pore across the TMD allows access of cytosolic acetyl-CoA to the luminal active site without being physically transported and exposed to the lysosomal lumen, protecting it from potential degradation. Furthermore, the pore is gated by subtle sidechain reorientation of a few residues, rather than going

through large conformational changes typical of membrane transporters[27]. This unique structure thus allows HGSNAT to catalyze the transmembrane acetylation reaction efficiently in one step.

The catalytic function of HGSNAT appears to be fulfilled by the TMD while the ECD and TM1 likely play regulatory roles. Consistent with this hypothesis, ECD and TM1 are only present in higher metazoans[4] and not well conserved (Supplementary Fig. 1). ECD likely facilitates optimal binding of long-chain HS in the lumen with its CBM, while the dynamics of TM1 could contribute to a mechanism that influences the binding/accessibility of acetyl-CoA from the cytosol. The auto-proteolytic site proposed to activate HGSNAT was identified in ICL1[11], which connects TM1 with the TMD. However, the identity of the protease, the exact cleavage site, and the mechanism by which it influences the activity of HGSNAT remain to be determined.

Although multiple strategies, including enzyme replacement therapy, gene therapy, chaperone therapy, and substrate reduction therapy are being explored for MPS[6], none are available to patients with MPS IIIC. The structure of HGSNAT provides the framework for understanding the mechanism of how the disease-causing variants lead to MPS IIIC at the molecular level. It also serves as the basis for rational drug design. For instance, the mutation hotspots suggest potential areas where folding and/or activity of HGSNAT can be manipulated and thus could be further explored, among other strategies.

## Methods
### Construct design

The cDNA encoding the open reading frame of human HGSNAT (Uniprot: Q68CP4-2) was ordered from GenScript and cloned into the pEG-BacMam expression plasmid[28]. Note that Q68CP4-2 corresponds to the short isoform 2 where 28 residues from the N-terminal are missing. However, we used the residue numbering for the canonical sequence (Q68CP4) throughout the paper. A tandem affinity purification tag composed of a twin ZpA963 tag and a 10xHis tag was fused to the C-terminal of HGSNAT separated by a thrombin cleavage site.

ZpA963 is an engineered affibody that binds to the Z variant of Protein A's B domain in high affinity[29]. Therefore, it can be used with commercial protein A resins for purification. The twin ZpA963 tag is designed to have a high binding capacity to facilitate purification.

## Protein expression, and purification

Recombinant human HGSNAT protein was expressed in HEK293 GnTI⁻ cells (ATCC). P2 virus was prepared in Sf9 cells and used to infect 1 L of HEK293 GnTI⁻ cells growing at 37 °C at MOI of 20–50. 24 hr post-infection, 10 mM sodium butyrate was added to the cells and the temperature was lowered to 30 °C. 48 hr post induction with sodium butyrate, the cells were harvested by centrifugation at 1000 x $g$ and stored at − 80 °C until use. The cell pellet was thawed on ice and resuspended in membrane resuspension buffer (50 mM Tris pH 7.5, 400 mM NaCl, 10% glycerol, 1 mM PMSF, 0.5 mM DTT, 20 μM DNase). 1% (w/v) $\beta$-dodecylmaltoside (DDM) (Anatrace) was added to solubilize the cell pellet with gentle stirring at 4 °C overnight. Insoluble materials were then removed the next day by centrifugation at 100,000 x $g$ at 4 °C for 30 min. The supernatant was incubated with the MabSelect SuRe resin (GE healthcare) for 4 h at 4 °C. The resin was then washed with 8 column volumes (CV) of wash buffer (20 mM HEPES pH 7.4, 150 mM NaCl, and 0.05% DDM). Bovine thrombin was added to the resin to remove the C-terminal tag with overnight on-column digestion. The flow-through fraction containing HGSNAT was collected and concentrated with Amicon 50 kDa MWCO filter to ~ 500 μL and further purified by size-exclusion chromatography (SEC) on a Superdex 200 increase 10/300 GL column in a buffer containing 20 mM HEPES pH 7.4, 150 mM NaCl, and 0.05% DDM (Supplementary Fig. 2c). Fractions corresponding to peak B are pooled and concentrated with Amicon 50 kDa MWCO filter and frozen in aliquots until further use.

## Cryo-EM sample preparation and data acquisition

Frozen aliquots of purified HGSNAT were subjected to final SEC in buffer containing 20 mM HEPES pH 7.4, 150 mM NaCl, and 0.01% glycodiosgenin (GDN) before preparing the sample for cryo-EM (Supplementary Fig. 2e). Peak B fractions were concentrated with 50 kDa MWCO concentrator (Amicon). For the Apo structure, concentrated HGSNAT at 2.9 mg/ml was used directly. For the complex samples, acetyl-CoA or CoA were added at final concentrations of 4.4 mM to the concentrated HGSNAT sample (3.1 mg/ml) and incubated on ice for 1 h. For the substrates complex and the products complexes, MU-$\beta$GlcN or MUF-NAG, in addition to acetyl-CoA or CoA, were added at final concentrations of 1.9 mM or 2.6 mM to the HGSNAT sample (4.4 mg/mL) and incubated on ice for 3 h. 3 μL of HGSNAT samples were applied to Quantifoil® holey carbon grids (Cu 1.2/1.3 300 mesh) glow-discharged with an PELCO easiGlow™ device using mixed air at 15 mA for 45 s. Grids were blotted with grade 595 standard Vitrobot filter paper for 4 s at 10 °C and 100% humidity using a Vitrobot® Mark IV, followed by rapid plunging into liquid ethane cooled by liquid nitrogen. The dataset for the Apo structures was collected at UCSF and the datasets for all other structures were collected at the Amgen/USC joint EM facility with parameters shown in Supplementary Table 1.

## Image processing

Micrographs were corrected for beam-induced drift using MotionCor2[30] outside cryoSPARC-v3 or using Patchmotion[31] within cryoSPARC. The contrast transfer function (CTF) parameters for each micrograph were determined using CTFFIND4.1[32] outside cryoSPARC or using PatchCTF[31] within cryoSPARC. Micrographs of poor quality were discarded. All subsequent processing was carried out in Relion 3.0[33] and cryoSPARC[34]. Similar strategies were used for all substrates/products bound structures as illustrated in Supplementary Fig. 3. Briefly, particles were selected from high-quality micrographs using templates previously generated. 2–3 rounds of 2D classification with 4x binned particles were used to remove clearly non-protein particles.

A subset of high-quality particles were picked based on 2D classification to generate an ab initio 3D model and used as a reference for subsequent 3D classification and refinement. Heterogenous refinement and non-uniform refinement were used iteratively to remove bad classes until no further improvement was observed. Global and local CTF refinement were then applied to further improve the resolution and map quality. The Apo dataset was processed initially in Relion 3.0, applying a strategy previously reported[35]. Briefly, after 2D classification, 3D classification was used to further classify the particle stack. Two different classes where TM1s in different conformations were observed and processed as separate classes (Apo$_{ground}$ and Apo$_{trans}$). Particle classes were then further processed by Relion auto-refine and Bayesian polishing strategy[36] and were imported into cryosparc using the pyem package v0.5[37]. Final maps obtained in both Relion and cryoSPARC achieved similar resolution and quality. Statistics of the final maps generated were calculated using the Fourier shell correlation (FSC) criterion and a threshold of 0.143 using the phenix_comprehensive_validation tool[38] and reported in Supplementary Table 1.

## Model building and structure refinement

A monomeric AlphaFold model of HGSNAT was docked into the final map of the Apo$_{ground}$ structure and used as the initial model. The dimeric model of the Apo$_{ground}$ structure was used as the initial model for all other structures. Unsharpened maps were used to dock and model the TM1 as they provided better connectivity of the backbone. Sharpened maps were used in all other areas in all structures. Local areas were adjusted and built with Coot-0.9.8 and first refined within Coot using real-space refinement with Ramachandran and geometry constraints. The full models were then refined in Phenix-1.20 using Phenix.real_space_refine without imposing constraints[38] against the sharpened maps. Figures were prepared with Pymol-3.5[39] and chimera-v1.7[40].

## Enzymatic assay

N-Acetyltransferase enzymatic activity of purified HGSNAT used for cryo-EM was measured using the fluorogenic substrate 4-methylumbelliferyl-$\beta$-d-glucosaminide (MU-$\beta$GlcN) (Pharmablock), as previously described[11] with slight modifications. Briefly, purified protein was added into a reaction mixture containing cell lysate generated by gentle sonication of non-transfected HEK293T cells in the presence of MU-$\beta$GlcN (Sigma-Aldrich, #69585) and acetyl-CoA (Sigma-Aldrich, #A2056). 25 μL reaction mixture containing 5 μL of 3 mM MU-$\beta$GlcN, 5 μL of 10 mM acetyl-CoA, 1 μL of purified HGSNAT at 5 mg/ml and 5 μL of cell homogenate is incubated at 37 °C for 3 h in McIlvain phosphate-citrate (0.2 M phosphate / 0.1 M citrate) buffer pH 5.5. The reaction was terminated by the addition of 975 μl 0.4 M glycine buffer, pH 10.4, and further diluted 50 times. 200 μL of diluted sample was transferred to a black 96-well plate and the fluorescence of released product $\beta$-Methylumbelliferone (4-MU) was measured in a BioTek Synergy Neo2 multimode microplate reader (excitation: 365 nm; emission: 445 nm). For the control sample, the same amount of HEK293T cell homogenate was added to the reaction without purified HGSNAT protein. The same assay was used to test the activities of different mutants using whole cell lysates of HEK293T cell transiently expressing HGSNAT-EGFP. Briefly, 10⁶ HEK293T cells were seeded in each well in a 6-well plate and grown for 24 h. Cells were then transfected with plasmid of HGSNAT in pEGFP-N1 vector at 2 μg DNA per well using X-tremeGENE™ HP DNA Transfection Reagent (Roche, 06366244001). Cells were harvested after 48 hrs and washed once with cold PBS buffer before being lysed with 200 μL IP Lysis Buffer (Thermo Scientific™, 87787) on ice for 10 min and centrifuged at 13,000 × $g$ for 10 min at 4 °C to remove cell debris. Total protein concentrations of the cell homogenates were determined by BCA Protein Assay Kit (Thermo Scientific™, 23227) according to the manufacturer's instruction. 25 μL assay mixture containing 5 μL of cell homogenate (2.8 μg/μL total protein concentration), 5 μL of 3 mM MU-$\beta$GlcN, 5 μL of 10 mM acetyl-CoA, and

5 μL of McIlvain phosphate-citrate buffer was incubated at 37 °C for 3 h. The reaction was terminated by the addition of 975 μL 0.4 M glycine buffer (pH 10.4). 200 μL of each reaction was transferred to a black 96-well plate and the fluorescence of 4-MU was read in a BioTek fluorometer same as for the purified protein (excitation: 365 nm; emission: 445 nm) (Supplementary Fig. 5a). Expression level of HGSNAT-EGFP wild-type and mutants were measured as GFP fluorescence level (Supplementary Fig. 5b) and used for normalization. Specifically, 10 μL cell homogenate (2.8 μg/μL total protein concentration) was diluted into 100 μL ddH$_2$O and measured by BioTek fluorometer (excitation: 485 nm; emission: 528 nm). Each assay was done several times with multiple technical replicates. Representative data were shown.

## Reporting summary

Further information on research design is available in the Nature Portfolio Reporting Summary linked to this article.

## Data availability

The atomic coordinates of HGSNAT structures have been deposited in the Protein Data Bank with the accession codes 8VLV (Apo$_{ground}$), 8VLY (Apo$_{trans}$), 8VKJ (Acetyl-CoA), 8VLG (Substrates complex), 8VLI (Products complex), and 8VLU (CoA). The corresponding maps have been deposited in the Electron Microscopy Data Bank with the accession codes EMD-43345 (Apo$_{ground}$), EMD-43348 (Apo$_{trans}$), EMD-43319 (Acetyl-CoA), EMD-43338 (Substrates complex), EMD-43339 (Products complex), and EMD-43344 (CoA). Source data are provided with this paper.

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

## Acknowledgements

We thank Dr. Eric Tse at UCSF Cryo-EM facility; Dr. Htet Khant at USC/Amgen joint Cryo-EM facility; Dr. Sara Weaver, Dr. Shengliu Wang, and Dr. Hui-Ting Chou at Amgen structural biology group for their support in data acquisition. We thank Hong Sun at Amgen Discovery Protein Science for her training and support in protein expression. We are grateful to Dr. Elena Ferri and Dr. Janet Finer-Moore for their critical reads of the manuscript. We thank the guidance provided by Dr. Ryan Potts and administrative support by Karen Walker at the Amgen R&D Postdoctoral Program. This work is supported by the Amgen R&D postdoc program. The initial investigation was partially supported by NIMH (K99MH119591) to F. L. Cryo-EM equipment at UCSF is partially supported by NIH grant S10OD020054 and Howard Huges Medical Institute. UCSF ChimeraX was developed with support from the National Institutes of Health R01-GM129325 and the Office of Cyber Infrastructure and Computational Biology, National Institute of Allergy and Infectious Diseases.

## Author contributions

F.L. conceived the project. Y. Zheng designed the ZpA963 tag. F.L., B.Z., P.N., A.B., and Y. Zheng expressed the protein. F.L, and B.Z. purified the protein, prepared samples for EM experiments, and acquired cryo-EM data. Z.C. performed the enzymatic assay with the guidance of Y. Zhou and F.L. F.L., and B.Z. determined the structures. F.L. analyzed the structures. R.M.S. and R.H.E. supervised the initial study. F.L., M.v.L.C. wrote the manuscript with input from all authors.

## Competing interests

B.Z., Z.C., Y. Zheng, Y. Zhou, M.v.L.C., and F.L. are employees of Amgen Inc., a for-profit organization. This study was conducted as a basic research project as part of Amgen's Postdoctoral Program project without commercial implications for Amgen Inc. The remaining authors declare no competing interests.
