## [Peer Review File · Nature Communications]

Structural and mechanistic insights into a lysosomal membrane enzyme HGSNAT involved in Sanfilippo syndromeREVIEWER COMMENTS

Reviewer #1 (Remarks to the Author):

This manuscript reports high-resolution cryo-EM structures of the lysosomal protein HGSNAT, an integral membrane enzyme that uses cytosolic Ac-CoA to acetylate the terminal glucosamine groups of heparan sulfate in the lysosomal lumen. This transmembrane acetylation is required to degrade heparan sulfate in lysosomes. Consequently, loss-of-function mutations of HGSNAT cause heparan sulfate accumulation and a severe lysosomal disease, Sanfilippo syndrome type C.

The authors show that HGSNAT forms a dimer with a novel structural fold. Each monomer comprises a beta-sandwich luminal domain linked to a long, tilted transmembrane helix (TM) and a compact membrane domain with 10 TMs. The luminal domain may serve to bind and position the heparan sulfate while the membrane domain catalyzes the enzymatic reaction. Interestingly, the authors report six high-resolution structures: 2 in apo states; 2 bound to one (acetyl-CoA) or both substrates (acetyl-CoA and a glucosaminide analog); and 2 bound to one (CoA) or both products. These structures provide detailed maps of HGSNAT along the reaction pathway, which elucidate the catalytic mechanism and help understand how the cytosolic and luminal substrates are brought together.

The study shows that a catalytic dyad H297-D307 triggers a nucleophilic attack of the sugar amine group onto the carbonyl group of acetyl-CoA, thus supporting the ternary complex mechanism rather than the ping-pong mechanism proposed by earlier biochemical studies. Another key finding is the existence of a long pore crossing the membrane domain, which accommodates the acetyl-CoA molecule and positions the acetyl moiety at the luminal mouth of the pore near the catalytic dyad and the bound glucosamine. Side chain rotation of residues lining the pore and a change in the tilt of TM1 would control access of cytosolic acetyl-CoA to the pore and the luminal active site. These findings are synthesized in a structural model of the transmembrane acetylation reaction. They also help predict how patient missense mutations impair HGSNAT structure and/or function.

The study is well designed and well executed. It unveils a novel structural fold and elucidates the mechanism of an atypical enzymatic reaction with a unique membrane topology. It will undoubtedly serve as a strong foundation for future HGSNAT studies in the context of Sanfilippo C disease and may help developing drugs that could rescue patient mutations.

Minor comments:

- line 141, '... acting as the nucleophile in both proposed mechanisms': a brief introduction to the two mechanisms and what distinguishes them would make it easier to follow the rest of the text.
- Figure 4: The lack of contrast (blue on blue) makes it impossible to see the CoA molecule in the pore.
- line 218, '... R372C/H, E499K and N286I directly disrupt the binding of substrates': Have the authors tested these mutants in their enzymatic assay or is there earlier experimental evidence for defective binding? If not, the sentence must be put in the conditional tense. There is a typo in 'substrates'.

Reviewer #2 (Remarks to the Author):

In this work, Zhao et al. reported six cryo-EM structures of HGSNAT, captured in the apo state, substrate-bound, and product-bound states. These structures provide insights into the reaction mechanisms of transmembrane acetylation mediated by HGSNAT. The structural data are robust. This work will be a useful addition to the field. I would recommend its publication provided that the following points are addressed:

1. A structure of HGSNAT was reported in Oct. 2023 on bioRxiv (<https://doi.org/10.1101/2023.10.23.563672>) and later in eLife (<https://doi.org/10.7554/eLife.93510.1>) by Navratna et al. It is very important to acknowledge the work from the other group. A proper discussion and structural comparison would seem appropriate.
2. The claims of novelty need to be toned down. See the list below:
 - Line 43: a "novel" structural fold. Considering the work of Navratna et al., these words need to be revised.
 - Lines 73-74: the words "novel" and "lack of any structural information" need to be revised.
 - Line 122: the binding site was identified at least partly by Navratna et al.
3. Line 96: Could the author elaborate on what exactly we learn from the [CoA] structure? It was not very clear in the manuscript.

4. Line 142: What calculations/measurements did the author perform to determine that the pKa of H297 has now become 4?

5. Performing some mutagenesis on the catalytic dyad would strengthen the story, but this is optional.

6. Line 186-196: Although I agree that the disulfide bond may not be important for dimerization, I would be cautious about drawing conclusions on intrinsic dynamics from comparing Apo inact and Apo trans. In Navratna et al.'s [AcetylCoA] structure, the disulfide bond is intact, but it appears to be broken in the current study, indicating some discrepancy. Furthermore, in the current study, the authors use 0.5 mM DTT in their first step of preparation, so the structures may not accurately reflect the native states of disulfide bond formation.

7. Lines 207-209: I am not sure one can draw that conclusion from the slight differences in the structures. What if both protomers are active but are just not synchronized, and you are capturing different time points of the reaction?

8. The Apo inact has a smaller pore than Apo trans, but how do the authors know it is inactive? An inactive state implies that the protein does not respond to substrates. However, it is possible that both apo states can accept substrates (maybe Apo inact has a lower affinity for example), and the binding of AcetylCoA triggers a conformational change to open the pore. This would change the model in Fig. 4.

Reviewer #3 (Remarks to the Author):

Li and coworkers report on the cryo-EM structures of heparan-a-glucosaminide N-acetyltransferase (HGSNAT) alone and bound to either substrates, products or reaction analogs. The structures reveal a dimer arrangement and a novel structural fold for an acetyltransferase. Based on the overall structure and comparisons, the authors propose a mode for how cytosolic acetyl-CoA is able to acetylate the luminal glucosamine and the structural changes that the enzyme undergoes to permit this. The authors also propose a catalytic ternary complex mechanism involving catalytic histidine and aspartic acid residues. The structure also allows the authors to rationalize the functional consequence of several disease mutations. Together, the studies are rigorous and largely compelling and clearly presented and provide an important advance in the field.

Some issues noted below should be addressed before publication:

1. Line 87, “and we observed no structure of the cleaved forms” is confusing. What I think the authors mean is that the fragments of HGSNAT-b could not be isolated on gel filtration.
2. Extended Data Figure 3, GSFSC and azimuth plots should be shown for all 6 structures.
3. Figure 2d, the residue labels in the circles are too small to read.
4. Figure 2e and 2f, I assume that the numbers indicated in red are hydrogen bond distances but what are the numbers indicated in parenthesis? This should be indicated in the legend.
5. Lines 144-157, The discussion of a ternary or ping-pong mechanism of catalysis that is presented in favor of a ternary complex mechanism seems reasonable but would be more definitive if the authors carried out multi-substrate kinetic analysis to distinguish between the two mechanisms.
6. Figure 3a, the authors should make clear in the legend the acetyl-CoA was removed from the structure shown.
7. Figure 3b,c, It is unclear which side chain conformations correspond to which structures. The side chains should be color-coded or described in some way in the legend. This is actually more clearly illustrated in Extended Data Figure 5.
8. Lines 178-193, I think that the argument that a disulfide bond between residues C362 of the dimer is weak as the residues are in position to form disulfide upon side chain rotation and the residues are strictly conserved (Extended Date Figure 1). Why can't the dimer still be dynamic in the presence of one disulfide bound?
9. Lines 207-210, it is unclear to me how the authors can conclude that the slightly different configuration of the active site suggests that only one monomer is active at a time. The authors also claim that this is consistent with publication 9 that they indicate reports different K_m values for acetyl-CoA and GlcN. The publication that is referenced, instead, reports two different K_m values for the GlnN substrate.

10. Line 218, “R372C/H, E499K and N286I directly disrupt the binding of substates GlcN and acetyl-CoA” should be reworded to “R372C/H, E499K and N286I are in position to directly disrupt the binding of substates GlcN and acetyl-CoA”

11. The methods section needs to be revised for grammar.

REVIEWER COMMENTS

Reviewer #1 (Remarks to the Author):

This manuscript reports high-resolution cryo-EM structures of the lysosomal protein HGSNAT, an integral membrane enzyme that uses cytosolic Ac-CoA to acetylate the terminal glucosamine groups of heparan sulfate in the lysosomal lumen. This transmembrane acetylation is required to degrade heparan sulfate in lysosomes. Consequently, loss-of-function mutations of HGSNAT cause heparan sulfate accumulation and a severe lysosomal disease, Sanfilippo syndrome type C.

The authors show that HGSNAT forms a dimer with a novel structural fold. Each monomer comprises a beta-sandwich luminal domain linked to a long, tilted transmembrane helix (TM) and a compact membrane domain with 10 TMs. The luminal domain may serve to bind and position the heparan sulfate while the membrane domain catalyzes the enzymatic reaction. Interestingly, the authors report six high-resolution structures: 2 in apo states; 2 bound to one (acetyl-CoA) or both substrates (acetyl-CoA and a glucosaminide analog); and 2 bound to one (CoA) or both products. These structures provide detailed maps of HGSNAT along the reaction pathway, which elucidate the catalytic mechanism and help understand how the cytosolic and luminal substrates are brought together.

The study shows that a catalytic dyad H297-D307 triggers a nucleophilic attack of the sugar amine group onto the carbonyl group of acetyl-CoA, thus supporting the ternary complex mechanism rather than the ping-pong mechanism proposed by earlier biochemical studies. Another key finding is the existence of a long pore crossing the membrane domain, which accommodates the acetyl-CoA molecule and positions the acetyl moiety at the luminal mouth of the pore near the catalytic dyad and the bound glucosamine. Side chain rotation of residues lining the pore and a change in the tilt of TM1 would control access of cytosolic acetyl-CoA to the pore and the luminal active site. These findings are synthesized in a structural model of the transmembrane acetylation reaction. They also help predict how patient missense mutations impair HGSNAT structure and/or function.

The study is well designed and well executed. It unveils a novel structural fold and elucidates the mechanism of an atypical enzymatic reaction with a unique membrane topology. It will undoubtedly serve as a strong foundation for future HGSNAT studies in the context of Sanfilippo C disease and may help developing drugs that could rescue patient mutations.

Minor comments:

- line 141, '... acting as the nucleophile in both proposed mechanisms': a brief introduction to the two mechanisms and what distinguishes them would make it easier to follow the rest of the text.

We thank the reviewer for the suggestion. We have added some background information and a schematic panel in figure 2 (Fig. 2e) to explain the two proposed mechanisms before discussion of our results. We also added experimental data (Fig. 2k, Extended Data Fig. 5) on the active site mutants to further support the analysis of the reaction mechanism. This section (line 176-183) and figure 2 have been modified.

- Figure 4: The lack of contrast (blue on blue) makes it impossible to see the CoA molecule in the pore.

Thanks for the suggestion. We have changed the color for CoA to red in figure 4 for better visualization. In an effort to keep consistent color scheme, we changed the color for figure 1 panel a as well.

- line 218, ‘... R372C/H, E499K and N286I directly disrupt the binding of substrates’: Have the authors tested these mutants in their enzymatic assay or is there earlier experimental evidence for defective binding? If not, the sentence must be put in the conditional tense. There is a typo in ‘substrates’.

Activities of R372C, R372H, E499K were tested in COS-7 cell expressing human HGSNAT and reported in *Feldhammer M, et al, 2009*. Activity of N286I was tested in COS-7 cell expressing HGSNAT in *Martins C, et al, 2018*. Both papers are cited now. However, the enzymatic assay does not test binding of substrate directly. We have therefore changed the sentence to “R372C/H, E499K and N286I are in position to directly disrupt the binding of substrates GlcN and acetyl-CoA and resulted in reduced activity when tested in vitro.” (Line 312-313). The typo has been corrected.

Reviewer #2 (Remarks to the Author):

In this work, Zhao et al. reported six cryo-EM structures of HGSNAT, captured in the apo state, substrate-bound, and product-bound states. These structures provide insights into the reaction mechanisms of transmembrane acetylation mediated by HGSNAT. The structural data are robust. This work will be a useful addition to the field. I would recommend its publication provided that the following points are addressed:

1. A structure of HGSNAT was reported in Oct. 2023 on bioRxiv (<https://doi.org/10.1101/2023.10.23.563672>) and later in eLife (<https://doi.org/10.7554/eLife.93510.1>) by Navratna et al. It is very important to acknowledge the work from the other group. A proper discussion and structural comparison would seem appropriate.

In general, the structure reported by *Navratna et al* (8TU9) is similar compared to the structures we report in this paper for the TM2-TM11 domain. However, TM1 is significantly different and 8TU9 adopts a conformation similar to the AlphaFold prediction. The construct used in *Navratna et al* has an N-terminal GFP tag retained in the final protein and the SDS-PAGE of *Navratna et al* did not contain the processed beta-subunit. While it is unclear if the conformation of TM1 observed in 8TU9 is influenced by the GFP tag and if it represents a different physiological state, observation of this conformation further confirms the dynamic nature of TM1. Orientation of monomer B relative to monomer A in 8TU9 is also different compared to the Acetyl-CoA bound structure in the current paper. This could be due to the different constructs as well as the purification method used in the two studies. These observations also highlight the flexibility of the dimer interface. We have now added discussion on these observations in the main text line

244-251 and included the structure reported by *Navratna et al* (8TU9) in Extended data fig. 8.

2. The claims of novelty need to be toned down. See the list below:

Line 43: a "novel" structural fold. Considering the work of Navratna et al., these words need to be revised.

Lines 73-74: the words "novel" and "lack of any structural information" need to be revised.

Line 122: the binding site was identified at least partly by Navratna et al.

We used the word "novel" mostly to refer to the structure itself. We have now change it to "unique" and "lack of any structural information until very recently" and cited the study of *Navratna et al.*

3. Line 96: Could the author elaborate on what exactly we learn from the [CoA] structure? It was not very clear in the manuscript.

We have added the following statements in line 124-129 to explain the rational for determining the [CoA] structure: *"Aside from being an integral part of the reaction cycle, the lack of extra density representing the acetyl group in the [CoA] structure compared to the [Acetyl-CoA] structure helped us to precisely model the position of the Acetyl group. Comparison of all 4 ligand bound structures helped us to better understand the acetyl transfer reaction."*

4. Line 142: What calculations/measurements did the author perform to determine that the pKa of H297 has now become 4?

We used ProPKA to calculate the pKa of the H297. We have added this information and reference in the manuscript, line 186.

5. Performing some mutagenesis on the catalytic dyad would strengthen the story, but this is optional.

We agree with the reviewer. We have added results from mutagenesis of residues H297, D307 and N286 to the manuscript (Fig. 2k, Extended Data Fig. 5, line 213-222).

6. Line 186-196: Although I agree that the disulfide bond may not be important for dimerization, I would be cautious about drawing conclusions on intrinsic dynamics from comparing Apo inact and Apo trans. In Navratna et al.'s [AcetylCoA] structure, the disulfide bond is intact, but it appears to be broken in the current study, indicating some discrepancy. Furthermore, in the current study, the authors use 0.5 mM DTT in their first step of preparation, so the structures may not accurately reflect the native states of disulfide bond formation.

The intrinsic dynamic of the dimer we referred to in the paper does not only apply to the [Apo inact] and [Apo trans] structures. For simplicity, only the [Acetyl-CoA], [Apo trans] and [Apo inact] structures are shown in figure 3 (all structures as well as 8TU9 now are included in Extended data Fig. 8f). Although we described the dimer interface in more details, the dynamic change is not restricted to the dimer interface. We concluded that the dimer is dynamic because of the large movement of the monomer B, as well as the dimer interface, between all 6 structures, as shown in Extended data Fig. 8. We have modified the text, line 260-261, to avoid confusion.

We understand the concern over the DTT used in the solubilization step. We added DTT in the solubilization step as our standard procedure to prevent non-specific aggregation and we have not observed it preventing disulfide bond formation under the low temperature and mild solubilization. In addition, the [Apo inact] and [Apo trans] structures were determined from the same cryo-EM dataset produced with the exact same protein preparation, suggesting that the presence of DTT in the very early step of the purification did not prevent the formation of the disulfide bond. Except for the [Apo inact] structure, the position of C362 in the other 5 structures in our study, including the [Acetyl-CoA] structure is compatible with disulfide bond formation. We only built disulfide bonds for the [Apo_{trans}] structure because the density is not sufficient in the other 4 structures. We think breaking of the disulfide is necessary during the transition between the very different conformations ECL2 adapts. However, the disulfide bond could reform after the transition, which might be the state *Navratna et al.*'s [Acetyl-CoA] structure captured. In fact, their structure appears to capture a slightly different dimer compared to our [Acetyl-CoA] structure (Extended data fig. 8f).

7. Lines 207-209: I am not sure one can draw that conclusion from the slight differences in the structures. What if both protomers are active but are just not synchronized, and you are capturing different time points of the reaction?

We agree with the reviewer. We did not mean to suggest the other monomer is not active. Just not in the active configuration at the exact same time. We have modified the text to "suggesting that the two monomers may be in active state asynchronously". Line 301-302.

8. The Apo inact has a smaller pore than Apo trans, but how do the authors know it is inactive? An inactive state implies that the protein does not respond to substrates. However, it is possible that both apo states can accept substrates (maybe Apo inact has a lower affinity for example), and the binding of AcetylCoA triggers a conformational change to open the pore. This would change the model in Fig. 4.

We agree with the reviewer that we do not know that the Apo_{inact} structure does not respond to substrate and do not mean to imply that. As we suggest in figure 4, the inactive and transition

state is in an equilibrium. We agree with the reviewer that the Apo_{inact} state may have a lower affinity less favorable for binding. We also agree that Acetyl-CoA could trigger a conformation change given that the pore is not completely open in the transition state. We think the transition state facilitate the binding of Acetyl-CoA and proceeding of the reaction. But it is possible for Acetyl-CoA to bind to the inactive state directly. However, given the observation of the Apo transition state, this alternative reaction path would be a minor path. To avoid confusion, we have changed the name of the inactive state to ground state throughout the paper.

Reviewer #3 (Remarks to the Author):

Li and coworkers report on the cryo-EM structures of heparan-a-glucosaminide N-acetyltransferase (HGSNAT) alone and bound to either substrates, products or reaction analogs. The structures reveal a dimer arrangement and a novel structural fold for an acetyltransferase. Based on the overall structure and comparisons, the authors propose a mode for how cytosolic acetyl-CoA is able to acetylate the luminal glucosamine and the structural changes that the enzyme undergoes to permit this. The authors also propose a catalytic ternary complex mechanism involving catalytic histidine and aspartic acid residues. The structure also allows the authors to rationalize the functional consequence of several disease mutations. Together, the studies are rigorous and largely compelling and clearly presented and provide an important advance in the field.

Some issues noted below should be addressed before publication:

1. Line 87, “and we observed no structure of the cleaved forms” is confusing. What I think the authors mean is that the fragments of HGSNAT-b could not be isolated on gel filtration.

Thanks for pointing this out. We indeed could not isolate the fragment of HGSNAT- β on gel filtration. But we meant to indicate that we did not solve a structure of only the TMD domain (TM2-TM11). If the two domains are separated after cleavage, we would expect to observe their individual structures. While the HGSNAT- α domain is most likely too small to be resolved by EM, we expect to be able to resolve the structure of the HGSNAT- β fragment where only the TMD (TM2-TM11) is present in the structure. We have modified the text to indicate the fragment is neither isolated in SEC nor observed in the EM experiment in line 103-104.

2. Extended Data Figure 3, GSFSC and azimuth plots should be shown for all 6 structures.

The GSFSC and azimuth plots for all structures are very similar. In the interests of space, we only showed one set of plots in the figure in the original draft. We agree with the reviewer that it could be helpful to show them for all reported structures. Unfortunately, we recently went through a group-wise effort to archive and remove data to save storage space. It would take significant effort to retrieve these plots at this point. We hope the PDB validation report and the deposited maps themselves are enough to demonstrate the quality and particle distribution of the EM data.

3. Figure 2d, the residue labels in the circles are too small to read.

We have made the residue labels larger.

4. Figure 2e and 2f, I assume that the numbers indicated in red are hydrogen bond distances but what are the numbers indicated in parenthesis? This should be indicated in the legend.

The numbers in parenthesis are the same distance in the other monomer. We have modified the figure legend to make it more clear now as *“Interactions between atoms are shown by dotted lines and distances (in Angstroms) between atoms involved are reported in red. Structure of only one monomer is shown but distances between the equivalent residues for the other monomer are labeled in parenthesis.”*

5. Lines 144-157, The discussion of a ternary or ping-pong mechanism of catalysis that is presented in favor of a ternary complex mechanism seems reasonable but would be more definitive if the authors carried out multi-substrate kinetic analysis to distinguish between the two mechanisms.

We thank the reviewer for the suggestion. Elegant kinetic analysis for HGSNAT had been reported in *Meikle et al*, 1995, where they showed that the Lineweaver-Burk plots clearly show converging sets of lines, indicating that the reaction proceeds via a ternary-complex mechanism. We have modified the text to clearly refer to this study. Line 192-193.

6. Figure 3a, the authors should make clear in the legend the acetyl-CoA was removed from the structure shown.

We have modified figure legend of Figure 3 to clarify that acetyl-CoA is not shown in all panels except panel d and e.

7. Figure 3b,c, It is unclear which side chain conformations correspond to which structures. The side chains should be color-coded or described in some way in the legend. This is actually more clearly illustrated in Extended Data Figure 5.

The sidechains are color-coded according to structures in panel a and the rest of Fig. 3. The small and transparency of the structure in panel b and c might makes it harder to see. We have added a description in the figure legend to help.

8. Lines 178-193, I think that the argument that a disulfide bond between residues C362 of the dimer is weak as the residues are in position to form disulfide upon side chain rotation and the residues are strictly conserved (Extended Data Figure 1). Why can't the dimer still be dynamic in the presence of one disulfide bound?

We agree with the reviewer the dimer can still be dynamic in the presence of the disulfide bond. We think the disulfide bond could break and reform during transitions between different states. Please refer to response to point #6 raised by reviewer 2 for more details.

9. Lines 207-210, it is unclear to me how the authors can conclude that the slightly different configuration of the active site suggests that only one monomer is active at a time. The authors also claim that this is consistent with publication 9 that they indicate reports different K_m values for acetyl-CoA and GlcN. The publication that is referenced, instead, reports two different K_m values for the GlnN substrate.

We concluded that only one monomer is active because the configuration of the active site in one monomer appears to be more compatible with the enzymatic reaction while the other is less so. For instance, in one monomer of the substrates complex (Extended Data Fig. 9i), the distance between the catalytic dyad D307 and H297 is 3.2 Å and the distance between N_{ϵ} of H297 and the N on glucosamine is 3.0 Å. These distances are compatible with bonding distances for the acetyl-transfer reaction. On the other hand, the same distances in the other monomer is 3.6 Å and 4.8 Å (Extended Data Fig. 9h), which are much further away and most likely not optimum for the catalysis. Therefore, we concluded that only one monomer is active in the dimeric structure we captured. As mentioned in our reply to point #7 raised by reviewer 2, we did not mean to suggest the other monomer is not active, just not at the same time. Slight conformation change of the active site would allow the “inactive” monomer to become active. We have modified the text to clarify. Line 300-302.

In table 4 of reference 9, Meikle *et al* reported 2 K_m for Acetyl-CoA, AcCoA (low) and AcCoA (high), as well as 2 K_m for GlcN (low), GlcN (High) and Disaccharide (low) and Disaccharide (high) at pH 7.0. Based on these data, the authors concluded that “the dual K_m values seen with the smaller monosaccharide substrate and the disaccharide substrate at pH 7.0 may be related to the accessibility of the substrates of the different enzyme forms. It would appear unlikely that there are more than one transferase present in the human lysosomal membrane, as all of the other enzymes involved in the degradation of heparin result from single gene products [25]. However, it is possible that the different K_m values result from differential processing of the enzyme. There are a number of reports of multiple forms of lysosomal enzymes, usually corresponding to the mature and precursor forms [26,27]. A second possibility may involve different conformational forms of the enzyme within the membrane....” Our structural observation is consistent with the possibility that there are different conformational forms of the enzyme where the active site configurations are not equally capable of catalysis.

10. Line 218, “R372C/H, E499K and N286I directly disrupt the binding of substrates GlcN and acetyl-CoA” should be reworded to “R372C/H, E499K and N286I are in position to directly disrupt the binding of substrates GlcN and acetyl-CoA”

We have changed this sentence.

11. The methods section needs to be revised for grammar.

We have revised the method section.

REVIEWERS' COMMENTS

Reviewer #2 (Remarks to the Author):

The authors have satisfactorily addressed my questions. I recommend the publication of this work.

Reviewer #3 (Remarks to the Author):

Li and coworkers have done a nice job addressing my concerns in the revised manuscript, which I think is now suitable for publication. The methods section could still use some grammatical corrections.